# Roles of WNT6 in Sheep Endometrial Epithelial Cell Cycle Progression and Uterine Glands Organogenesis

**DOI:** 10.3390/vetsci8120316

**Published:** 2021-12-09

**Authors:** Xiaoxiao Gao, Xiaolei Yao, Xiaodan Li, Yaxu Liang, Zifei Liu, Zhibo Wang, Kang Li, Yingqi Li, Guomin Zhang, Feng Wang

**Affiliations:** 1Jiangsu Livestock Embryo Engineering Laboratory, College of Animal Science and Technology, Nanjing Agricultural University, Nanjing 210095, China; 2017205002@njau.edu.cn (X.G.); yaoxiaolei@njau.edu.cn (X.Y.); 2019205003@njau.edu.cn (X.L.); 2018105026@njau.edu.cn (Y.L.); 2019205002@njau.edu.cn (Z.L.); 2018105081@njau.edu.cn (Z.W.); 2020205015@stu.njau.edu.cn (K.L.); 2019805113@stu.njau.edu.cn (Y.L.); zhangguomin@njau.edu.cn (G.Z.); 2Hu Sheep Academy, Nanjing Agricultural University, Nanjing 210095, China

**Keywords:** WNT6, cell cycle, endometrial epithelial cell, organoid, Hu sheep

## Abstract

The uterus, as part of the female reproductive tract, is essential for embryo survival and in the maintenance of multiple pregnancies in domestic animals. This study was conducted to investigate the effects of WNT6 on Hu sheep endometrial epithelial cells (EECs) and uterine glands (UGs) in Hu sheep, with high prolificacy rates. In the present study, Hu sheep with different fecundity, over three consecutive pregnancies, were divided into two groups: high prolificacy rate group (HP, litter size = 3) and low prolificacy rate group (LP, litter size = 1). A comparative analysis of the endometrial morphology was performed by immunofluorescence. RNA-seq was used to analyze the gene’s expression in endometrium of HP and LP Hu sheep, providing a candidate gene, which was investigated in EECs and organoid culture. Firstly, higher density of UGs was found in the HP Hu sheep groups (*p* < 0.05). The RNA-seq data revealed the importance of the WNT signaling pathway and WNT6 gene in Hu sheep endometrium. Functionally, WNT6 could promote the cell cycle progression of EECs via WNT/β-catenin signal and enhance UGs organogenesis. Taken together, WNT6 is a crucial regulator for sheep endometrial development; this finding may offer a new insight into understanding the regulatory mechanism of sheep prolificacy.

## 1. Introduction 

Hu sheep are considered a prolific breed. As in other prolific breeds, this trait is associated with a particular genetic background (related to the Booroola gene, *FecB*). Interestingly, some ewes carrying the *FecB* gene, a marker of a high ovulation rate, still reveal a low prolificacy rate [1]. Thus, differences in the fecundity of sheep would derive, likely, from non-genetic influences (such as nutrition). Mammalian fecundity is controlled by ovulation rate and uterine receptivity. Uterine development and receptivity are closely related to the prolificacy rate of Hu sheep [2]. Even though many studies have focused on the regulation of ovulation, little research has been conducted on the effects of uterine development on fertility. As previously reported, uterine glands knocked out (UGKO) ewes exhibited sterility, which was attributed to recurrent pregnancy loss as a result of the absence of specific embryotrophic secretions of uterine glands [3,4,5]. 

Uterine adenogenesis induced by various growth factors, such as WNTs, FGFs, and IGFs [2,6]. Furthermore, WNTs play an important role in mouse uterine development, regulating the receptivity of the uterus to embryo implantation [6]. In human and mice models, the WNT family encodes 19 highly conserved secretory glycoproteins that are involved in gland morphogenesis in the breast and kidney by regulating cell growth and differentiation [7,8,9]. Many members of the WNTs family are expressed in both the proliferative and secretory phases of the human endometrium [10]. In newborn sheep, WNT2B was mainly expressed in the uterine stroma, while WNT5a, WNT7a, and WNT11 were mainly expressed in the uterine epithelium [9]. The canonical WNT signaling pathway is associated with trophoblast differentiation and tubular organ differentiation, in humans [11,12] and sheep [13]. A previous study showed that WNTs promote the proliferation, migration, and invasion of esophageal cancer cells by governing the expression of their target genes, such as Myc and Cyclin D1 [14]. Nevertheless, the role of WNT6 in Hu sheep endometrial stromal cells has not been investigated.

As already reported, traditional two-dimensional (2D) cell cultures cannot faithfully represent the in vivo biology of tissues and organs in the body. Meanwhile, over the past decade, 3D cell culture techniques have been developed as a powerful tool for studying tissue biology and disease models [15]. Therefore, this study explored the roles of WNT6, as a candidate gene in the development of sheep endometrium at the cellular and organoid levels, to provide a theoretical reference for elucidating the developmental mechanism of uterine glands (UGs) in Hu sheep.

## 2. Materials and Methods 

This study was carried out according to the Guide for the Care and Use of Laboratory Animals (SYXK 2011-0036) prepared by the Ethics Committee of Nanjing Agricultural University.

### 2.1. Preparation of Animals and Tissues

A total of 6 healthy pluriparous ewes (3 months after lambing, average age: 2.5 ± 0.2 years) were selected from the nucleus herds of Hu sheep at Taizhou Hailun Sheep Industry Co., Ltd. (Taizhou, China) in September. The selected Hu sheep were assigned to a high prolificacy rate group (HP, *n* = 3, litter size = 3) and a low prolificacy rate group (LP, *n* = 3, litter size = 1) according to their records of littering, and had free access to food and water under natural lighting. The estrus of sheep was synchronized by intra-vaginal progestagen sponges (30 mg; Ningbo Sansheng pharmaceutical Co., LTD, Zhejiang, China) for 11 days, and 100 IU of prostaglandin-F2α (Ningbo Sansheng Co., Ltd.) was given at sponge removal as previously described [16]. The sheep were slaughtered at the second estrus (natural estrous), and the mid-part of the left uterine horn was immediately collected and divided into two portions. The endometrium was scraped from one portion, and immediately stored at −80 °C for RNA and protein extraction. Another sample was fixed with 4% paraformaldehyde for immunofluorescence [17].

### 2.2. Immunofluorescence 

Immunofluorescence of uterine tissue sections was performed as previously described with minor modifications [18]. Paraffin-embedded samples were cut into 5-mm thick sections deparaffinized in xylene and rehydrated using an ethanol gradient. Antigen retrieval was carried out in citrate buffer solution at 100 °C for 5 min. After cooling down to room temperature, the sections were blocked with 5% bovine serum albumin (BSA) at 37 °C for 30 min. Then sections were incubated overnight at 4℃ with the primary antibody (Table 1). Sections were then washed with PBS and incubated with corresponding secondary antibody (A0453, 1:1000 dilution, Beyotime, Shanghai, China) at room temperature for 1 h. The negative control was not incubated with primary antibody, and was only treated with secondary antibody. FOXA2, as a UG-associated antigenic marker, is widely used to assess the development of UGs of mammalian uteri [19,20]. In this study, UGs were stained with a FOXA2 antibody (Table 1) to assess the density of UGs in different groups. Sections were taken to calculate the UGs by a Nikon microscope (Nikon Inc., Tokyo, Japan) under the 555 nm excitation wavelength. The density of UGs was counted at high magnification in each of the five random sampling areas in sections [17,21].

### 2.3. RNA-Seq 

The total RNA of sheep endometrium tissues was isolated by TRIzol (Invitrogen, Carlsbad, CA, USA) after homogenization. RNA quality and concentrations were examined by the RNA Nano 6000 Assay Kit of the Agilent Bioanalyzer 2100 System (Agilent Technologies, Santa Clara, CA, USA). Sequencing libraries were generated using the NEB Next Ultra Directional RNA Library Prep Kit for Illumina (NEB, Ipswich, MA, USA), and index codes were added to attribute sequences to each sample. RNA sequencing was conducted on Illumina HiSeq 4000 platform (Illumina, San Diego, CA, USA), and 150 bp paired-end reads were generated. We obtained clean reads after removing low-quality reads of each sample, and the final clean reads were then mapped to the Ovis aries reference genome (Oar v4.0) using HiSAT2 [22]. 

### 2.4. Differential Expression and Functional Enrichment Analysis 

The expression levels of mRNAs in each sample were calculated by FPKM (fragments per kilobase of exon per million fragments mapped) [23]. The statistically significant differentially expressed (DE) genes were identified by a *p*-value threshold of < 0.05 and |log2(fold change)| > 1, and a hierarchical clustering analysis was performed by the R package DESeq (the R Foundation, Vienna, Austria) according to the FPKM of different groups [24]. In this study, KEGG (http://www.genome.jp/kegg. 5 June 2021) pathway analysis was performed by the KOBAS software [25,26]. Gene ontology (GO) enrichment analysis was implemented by the topGO R packages, and corrected by *p*-value [27]. 

### 2.5. RT-qPCR

The RT-qPCR analysis was utilized to validate RNA-seq data. We used Primer 5 (Premier Biosoft, Palo Alto, CA, USA) software to design primers online, and assessed via the Basic Local Alignment Search Tool (BLAST; National Center for Biotechnology Information [NCBI], Bethesda, MD, USA). ChamQ Universal SYBR qPCR Master Mix (Vazyme, Nanjing, China) was used for RT-qPCR. Blank control included water instead of cDNA. The expression levels of mRNA were evaluated by 2^−ΔΔCT^, and normalized to the housekeeping gene *ACTB* expression. All used primers are shown in Table 2.

### 2.6. Western Blot Analysis 

Western blot analysis was performed according to our previously described procedure with minor modifications [28]. For total protein extraction, each tissue sample was homogenized in radioimmunoprecipitation (RIPA) lysis buffer (Beyotime, Shanghai, China), and then quantified by bicinchoninic acid assay (BCA) method. The protein samples were separated in a 12% sodium dodecyl sulfate polyacrylamide gel electrophoresis (Invitrogen, Shanghai, China), and then electro-transferred onto polyvinylidene fluoride (PVDF) membranes (Millipore, Billerica, MA, USA). The membrane was incubated with a primary antibody (Table 1) overnight at 4 °C, after washing in Tris-buffered saline with Tween 20 (TBST), and subsequently incubated with a secondary antibody (1:5000, Goat anti-Rabbit IgG; Thermo Fisher Scientific, Waltham, MA, USA) for 1 h. Protein signals were detected using an enhanced chemiluminescence western blot detection system (Fujifilm, Tokyo, Japan).

### 2.7. Transient Transfection in Endometrial Epithelial Cells (EECs)

EECs were isolated and identified from sheep according to our previous study [28]. Briefly, the epithelial layer from fresh uterus of sheep was scraped off using a surgical knife blade, and washed by DMEM/F12 (Invitrogen, Carlsbad, CA, USA). Then the tissues were finely minced by scissors into fragments (<1 mm^3^), and plated onto 10-cm plates in DMEM/F12 containing 20% FBS. After 5 days of culture, we obtained EECs by differential centrifugation. The overexpression vector (pEX-4-WNT6) and small interfering RNA (si-WNT6) were synthesized by GenePharma (Shanghai, China). EECs transfected with the empty vector (pEX-4) and siRNA with a scrambled sequence served as negative controls (pEX-4-NC and NC respectively). When the EEC were 70% confluent in 6 well plates, the pEX-4-WNT6 plasmid or si-WNT6 were transfected into sheep EECs using Lipofectamine 3000 (Invitrogen, Shanghai, China) [29,30]. After 48 h transfection, the transfected cells were collected for total RNA extraction and protein analysis.

### 2.8. Analysis of Cell Cycle 

Cell cycle analysis was carried out using a cell cycle staining kit (KeyGEN BioTECH, Nanjing, China) according to the manufacturer’s instructions. After 48 h transfection, EECs were fixed by cold 70% ethanol, and the incubated with propidium iodide (KeyGEN BioTECH) and RNase A for 30 min. Flow cytometry (FACSCalibur; Becton, Dickinson and Company, Franklin Lakes, NJ, USA) was used to assess the cell cycle, the percentage of cell distribution in G0/G1, S, and G2/M phase was measured, and the data were analyzed by CELL Quest software [31].

### 2.9. Organoid Culture of Ovine UGs

EECs were cultured to 80% confluence and then digested into single cells with 0.25% trypsin. The density of EECs was adjusted to 1 × 10^5^/mL and then inoculated into the Transwell plates containing Matrigel glue and 300 μL of complete medium was added in each well. The growth of organs was observed regularly, and identified by cytokeratin and FOXA2 immunofluorescence staining [19,20]. Organoids were obtained from Matrigel glue by a cell recovery solution, and the expression of genes related to glandular development was detected by RT-qPCR.

### 2.10. Statistical Analysis

SPSS 19.0 software (Statistical Package for the Social Sciences, Chicago, IL, USA) was used to analyze the data. All data were tested for normality. Expression level of DE genes was calculated using a *t*-test, which was presented as the means ± standard deviations (SDs). The significance level of the data were defined as *p* < 0.05.

## 3. Results

### 3.1. Analysis of UGs in Hu Sheep with Different Fecundity

As shown in Figure 1A,B, the density of UGs in the HP group of Hu sheep was higher than that in the LP group (*p* < 0.05). Furthermore, the gene expression levels of growth factors related to UG development were detected by RT-qPCR, and results showed that the expression levels of *HOXA10*, *HOXA11*, *FOXA2*, *PLGF*, *IGF-1*, *PGR*, *VEGFA*, and *LGR5* mRNA in the HP group were higher than those in the LP group (*p* < 0.05; Figure 1C). 

### 3.2. Analysis of DE mRNAs in Endometrium by RNA-Seq

RNA-seq analysis was performed to screen DE mRNAs from the endometrium of Hu sheep with different fecundity. In total 874 DE mRNAs were identified in endometrium between the HP and LP groups (Appendix A), with 440 upregulated and 434 downregulated DE mRNAs (*p* < 0.05; Figure 2A). Clustering analysis of DE mRNAs showed that genes related to uterine function were highly expressed in endometrium of the HP group, such as *HOXA11*, *WNT6*, and *PLGF* (Figure 2B). Furthermore, KEGG pathway analysis and Gene Set Enrichment Analysis (GSEA) discovered that WNT signaling pathway was significantly enriched in the HP group (Appendix A; Figure 2C,D). The expression level of DE mRNAs in WNT signaling pathway was confirmed by RT-qPCR; *WNT6* was highly expressed in endometrium of the HP group than that in the LP group (*p* < 0.05; Figure 2E).

### 3.3. Effects of WNT6 on β-Catenin Expression in EECs

The fragment of *WNT6* cDNA was obtained from the endometrium of Hu sheep, and sequence alignment of WNT6 homologous amino acids was analyzed to various species, results indicated ovine WNT6 had high homology with goat (99.73%), cattle (99.45%), pig (97.26%), and human (97.81%) WNT6 amino acid sequences (Appendix A). 

The function of WNT6 was investigated by WNT6 suppressing and overexpressing in EECs. Results indicated that knockdown of WNT6 significantly inhibited the expression level of β-catenin (*p* < 0.05; Figure 3A,B), the downstream effector of WNT pathway. While overexpression of WNT6 significantly increased the level of β-catenin (*p* < 0.05; Figure 3C,D). 

### 3.4. The Effect of WNT6 Transfection on Cell Cycle of EECs 

As compared with the control groups, WNT6 knockdown in EECs revealed an apparent G0/G1-phase cell cycle arrest (*p* < 0.05; Figure 4A,B), whereas WNT6 overexpression sharply increased the percentage of cells in S phase, with a marked decrease of cells in G0/G1 phase (*p* < 0.05; Figure 4C,D). Results also indicated that WNT6 knockdown significantly inhibited the expression level of PCNA and c-MYC proteins, as cell cycle-related factors; while WNT6 overexpression significantly increased the expression level of PCNA, c-MYC, and cyclin D1 proteins (*p* < 0.05; Figure 4E,F). 

### 3.5. Effects of WNT6 on Organoid Development of Ovine UGs

Results of organoid culture in a 3D environment showed that spherical organs appeared on the seventh day (Figure 5A). Furthermore, endometrial glands from organoids expressed the FOXA2 as a UGs marker (Figure 5B). 

In a 3D environment, WNT6 knockdown inhibited the organoids formation, whereas WNT6 overexpression had the opposite effect (*p* < 0.05; Figure 6A,B). Furthermore, RT-qPCR results showed that WNT6 knockdown inhibited the expression of glandular development related genes, such as *FGF7*, *FGF1*0, *HOXA10*, *LGR5,* and *PRLR,* Whereas WNT6 overexpression significantly promoted the expression of *FGF7*, *FGF10*, *LGR5*, *LIF*, and *PRLR* in organoid culture (*p* < 0.05; Figure 6C,D).

## 4. Discussion

Although a high ovulation rate gene, such as *FecB,* may be used to select sheep with large litter size productivity [2], uterine capacity is also critical for embryo survival. Furthermore, the endometrial gland is an essential uterine structure for reproduction in mammals. Recent studies of sheep and mice have established that uterine gland products and secretions are crucial for normal uterine functions, such as uterine receptivity, decidualization, embryo implantation, conceptus survival and growth, and placental development [32,33]. The UGKO ewes were completely infertile with recurrent miscarriages [34]. As noted in pigs, increasing uterine horn length, and/or endometrial gland number may be a rational approach to decreasing early embryonic loss and increasing litter size [2,35]. Therefore, the development of endometrial glands is essential for uterine capacity to maintain large litter size. Interestingly, we discovered a higher density of UGs in high prolificacy rate Hu sheep, which may be beneficial to multiple pregnancies of Hu sheep. As previously reported, the postnatal uterine adenogenesis is intrinsically regulated by several growth factors including FGFs, IGFs, VEGF, FOX, HOXA, and hepatocyte growth factor [33]. Furthermore, PLGF secreted from human UGs modulates the endometrial functions, and supports the implantation of conceptus [36], and LGR5 is essential for mice uterine gland development [37]. In the present study, these growth factors were highly expressed in endometrium of Hu sheep with high prolificacy rate. These findings highlight the significance of UG development in Hu sheep with high prolificacy rate.

In order to study the further mechanism of UG development in high prolificacy sheep, RNA-seq was performed to screen the DE mRNAs involved in uterine functions. Previous studies reported that HOXA10, WNTs, and PLGF were essential for successful embryo implantation in mammals [33,36,38]. In this study, DE mRNAs *HOXA11*, *WNT6*, and *PLGF* were highly expressed in HP sheep, which may explain the high density of UGs in high prolificacy Hu sheep. Our bioinformatics analysis showed that the DE mRNAs were significantly enriched in TGF-β and WNT pathways. In the past two decades, various components of WNT signaling have been found in mouse [39], sheep [9], and human [10] uterus. Roles of WNT signaling have been clarified in UG formation and normal implantation in the mouse model [6]. These findings suggest that the activity of WNT signaling may play crucial roles in endometrial adenogenesis. In this study, we demonstrated a higher level of WNT6 expression in high prolificacy rate Hu sheep. Previous research showed that WNT6, as a candidate for the ureter epithelium-derived signal, was involved in the activation of kidney development [40]. However, the role of WNT6 in the UG development of high prolificacy Hu sheep has not been discovered. In the present study, a high homology of the sheep WNT6 amino acids was found to that of other livestock and human. 

Our previous study discovered WNT6 was mainly expressed in EECs of sheep [41]. To clarify the mechanism of WNT6 in the regulation of uterine epithelial growth, we isolated and cultured sheep EECs for transfection experiments. In the canonical WNT/β-catenin signaling pathway, β-catenin as the downstream effector of WNTs, activates the transcription of target genes and regulates endometrial cell proliferation [42,43]. In the present study, WNT6 promoted the expression of β-catenin in EECs, which was consistent with previous research. Endometrial adenogenesis is a primary event for survival and development of the conceptus in laboratory and domestic animals [2]. The formation of UG is based on the epithelial population growth by EECs proliferation [44]. Furthermore, our results indicated that WNT6 could enhance the process of the cell cycle of EECs, which is consistent with previous studies that WNT signaling is a direct link with the cell cycle machinery [45]. 

Traditional 2D cell cultures, in general, do not accurately represent the UGs biology in vivo, and it is difficult to maintain its phenotype and hormonal responsiveness [15]. Meanwhile, organoid culture has a superior ability of tissue expansion, retains the original phenotype, and is closer to the development state of endometrial glands in vivo [15,46]. Therefore, organoid culture was performed herein and identified by FOXA2 as a marker of endometrial glands [20]. Results of our transfection test, at the organoid level, showed that WNT6 regulated endometrial epithelial organogenesis, and the expression of organogenesis related genes, indicating that WNT6 may be an important regulator of UG development. 

## 5. Conclusions

In this study, we demonstrated that the density of UGs in Hu sheep with high prolificacy rate was superior to that in low prolificacy rate Hu sheep. Our systematical analysis of RNA-seq data revealed DE mRNAs related to UG development. Importantly, our study indicates that WNT6 enhances sheep cell cycle progression of EECs and UG organogenesis (Appendix A). Overall, this research offers new insight into understanding the regulatory mechanism of UG development in high prolificacy sheep. 

## Figures and Tables

**Figure 1 vetsci-08-00316-f001:**
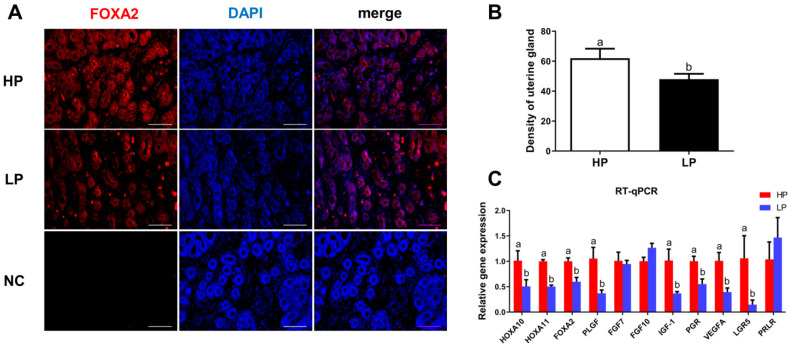
Analysis of UGs in Hu sheep with different fecundity. (**A**) Immunolocalization of UGs using FOXA2 antibody. UGs, uterine glands. HP, high prolificacy rate Hu sheep; LP, low prolificacy rate Hu sheep; NC, negative control was not incubated with primary antibody; DAPI, 4, 6-diamidino-2-phenylindole, blue nuclear stain. (**B**) Analysis the density of the UGs. (**C**) Relative expression of genes involved in UG development. HOXA10/11, Homeobox A10/11; FOXA2; Forkhead box A2; PLGF, placental growth factor; FGF7/10, Fibroblast growth factor 7/10; IGF-1, insulin like growth factor 1; PGR, progesterone receptor; VEGFA, vascular endothelial growth factor A; LGR5, leucine rich repeat containing G protein-coupled receptor 5; PRLR, prolactin receptor. The values represent means ± SDs. Means with different letters indicate significant differences (*p* < 0.05). Bar = 50 μm.

**Figure 2 vetsci-08-00316-f002:**
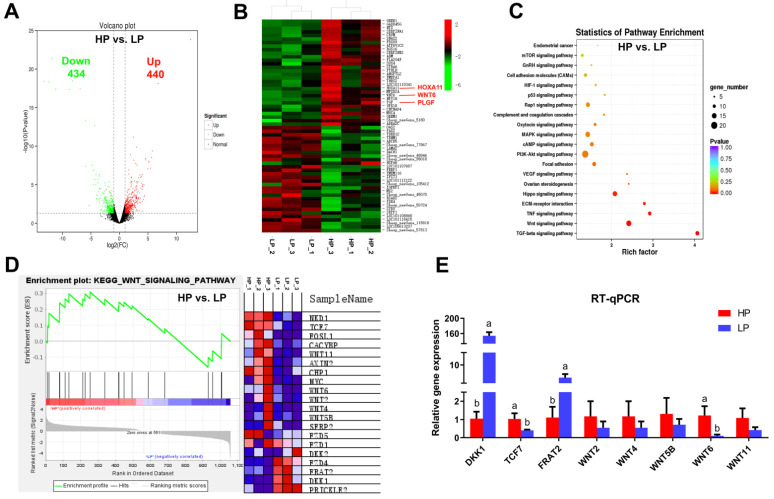
Analysis of DE mRNAs in the endometrium of Hu sheep with different fecundity. (**A**) Volcano plots of DE mRNAs. (**B**) The clustering analysis of mRNAs that were differentially expressed analysis of DE mRNAs (Top 30). (**C**) KEGG functional annotation of DE mRNAs. (**D**) Overview of GSEA used to identify the differential gene profiles in WNT signaling pathway. (**E**) Validation of RNA-seq results by RT-qPCR. The values represent means ± SDs. Means with different letters indicate significant differences (*p* < 0.05).

**Figure 3 vetsci-08-00316-f003:**
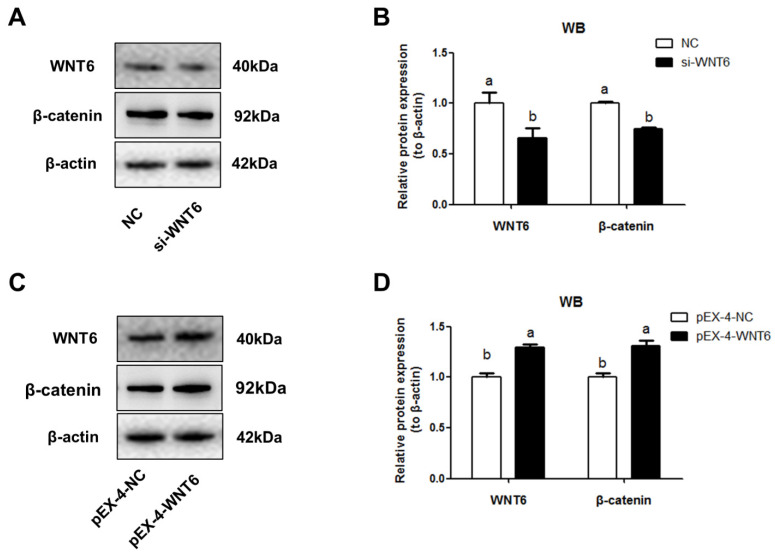
The effects of WNT6 transfection on β-catenin expression in EECs. (**A**,**B**) Relative expression of WNT6 and β-catenin protein to β-actin by WNT6 knockdown. (**C**,**D**) Relative expression of WNT6 and β-catenin protein to β-actin by WNT6 overexpression. si-WNT6, the small interfering RNA for *WNT6*. pEX-4-WNT6, pEX-4 vector with *WNT6*-CDs inserted. siRNA with scrambled sequence absent in the sheep genome and pEX-4 was severed as negative control for knockdown and overexpression, respectively. The values are presented as means ± SDs. Means with different letters indicate significant differences (*p* < 0.05).

**Figure 4 vetsci-08-00316-f004:**
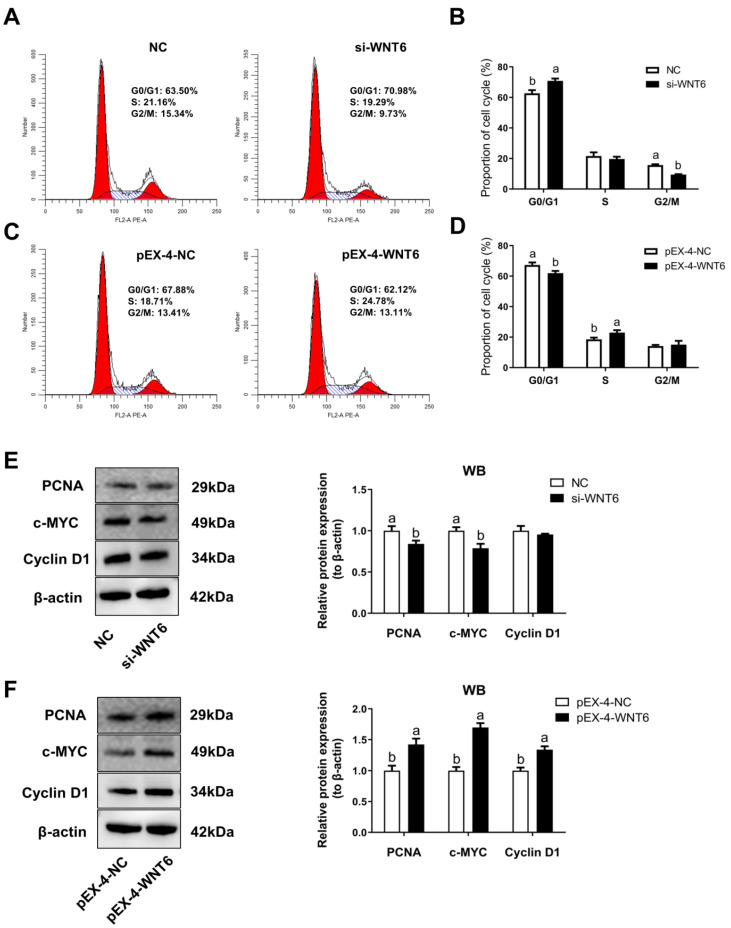
The effect of WNT6 transfection on cell cycle of EECs. (**A**,**C**) Analysis of cell cycle by flow cytometry. (**B**,**D**) The percentages of G0/G1, S, and G2/M phases of the cell cycle. (**E**,**F**) Effects of WNT6 on protein expression related cycle of EECs. NC, Negative control was the siRNA with scrambled sequence absent in the sheep genome. si-WNT6, the small interfering RNA for *WNT6*. pEX-4-NC, Negative control of pEX-4 was the empty vector. pEX-4-WNT6, pEX-4 vector with *WNT6*-CDs inserted. Values are presented as the mean ± SDs. Means with different letters indicate significant differences (*p* < 0.05).

**Figure 5 vetsci-08-00316-f005:**
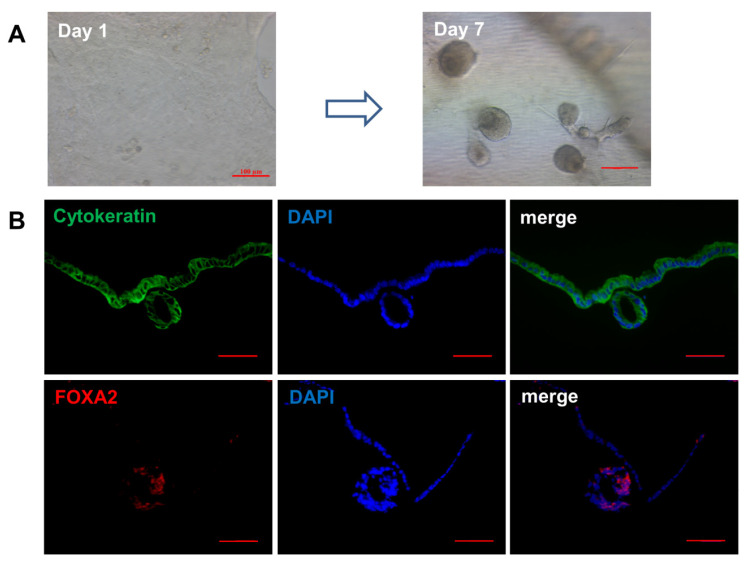
The formation and identification of ovine UGs organoids. (**A**) The formation of UGs organoids in Matrigel. (**B**) The identification of organoids by cytokeratin (green) and FOXA2 (red) immunoreactivity. Bar = 100 μm.

**Figure 6 vetsci-08-00316-f006:**
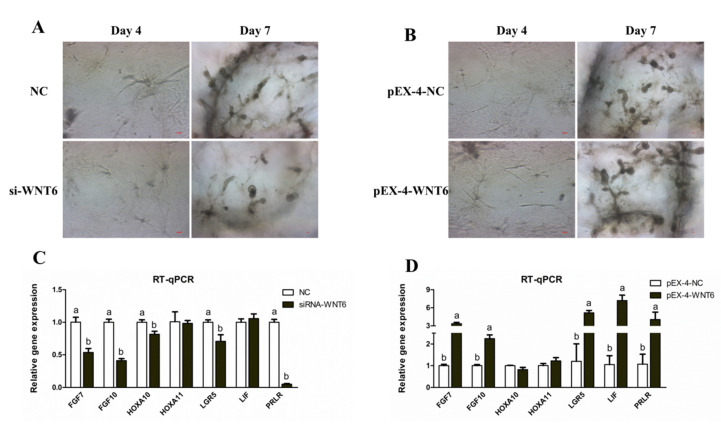
Effects of WNT6 transfection on the development of ovine UGs organoid. (**A**) Effects of WNT6 suppression (si-WNT6) on organoid development. (**B**) Effects of WNT6 overexpression (pEX-4-WNT6) on organoid development. (**C**) The expression of glandular development related genes in UGs organoid by WNT6 suppression. (**D**) The expression of glandular development related genes in UGs organoid by WNT6 overexpression. NC, Negative control was the siRNA with scrambled sequence absent in the sheep genome. si-WNT6, the small interfering RNA for *WNT6*. pEX-4-NC, Negative control of pEX-4 was the empty vector. pEX-4-WNT6, pEX-4 vector with *WNT6*-CDs inserted. Values are presented as the mean ± SDs. Means with different letters indicate significant differences (*p* < 0.05). Bar = 50 μm.

**Table 1 vetsci-08-00316-t001:** Details of antibodies.

Antibodies	Cat No.	Source	IF	WB	Definition/Function
PCNA	ab15497	Abcam, Cambridge, UK	_	1:600	Proliferating cell nuclear antigen, commonly known as PCNA, is a protein that acts as a processivity factor for DNA polymerase δ in eukaryotic cells.
WNT6	ab154144	Abcam, Cambridge, USA	_	1:1000	The WNT gene family consists of structurally related genes, which encode secreted signaling proteins.
β-catenin	51067-2-AP	Proteintech, Wuhan, China	_	1:5000	β-Catenin, also known as CTNNB1, is a key downstream component of the canonical Wnt pathway that plays diverse and critical roles in embryonic development and adult tissue homeostasis.
Cyclin D1	26939-1-AP	Proteintech, Wuhan, China	_	1:1000	Cyclin D1, also known as PRAD1 or BCL1, belongs to the highly conserved cyclin family, whose members are characterized by a dramatic periodicity in protein abundance throughout the cell cycle.
c-MYC	10828-1-AP	Proteintech, Wuhan, China	_	1:1000	MYC is a multifunctional, nuclear phosphoprotein that plays a role in cell cycle progression, apoptosis, and cellular transformation.
FOXA2	Ab108422	Abcam, Cambridge, USA	1:300	_	FOXA2 as a transcription factor involves in embryonic development, establishment of tissue-specific gene expression and regulation of gene expression in differentiated tissues, and it is a mark of epithelial cell
Cytokeratin	AF1618	Beyotime, Shanghai, China	1:100	_	Cytokeratin are a large family of proteins that form the intermediate filament cytoskeleton of epithelial cells.
β-actin	bs-0061R	Bioss, Beijing, China	_	1:2000	Actins are highly conserved globular proteins that are involved in various types of cell motility and are ubiquitously expressed in all eukaryotic cells. β-actin is the protein for internal reference.

IF, immunofluorescence; WB, Western blot; PCNA, proliferating cell nuclear antigen; WNT6, Wnt family member 6; c-MYC, MYC proto-oncogene; FOXA2, forkhead box A2. The sequence homology of the target sequence for PCNA, WNT6, β-catenin, Cyclin D1, c-MYC, FOXA2, and Cytokeratin was more than 85% with the antibodies cross-reactive species.

**Table 2 vetsci-08-00316-t002:** Primers and sizes of the amplification products of tested genes.

Gene	Forward Primer, 5′-3′	Reverse Primer, 5′-3′	Product Size (bp)
*HOXA10*	CTTTTCGCGTCCAGAGACCC	ATCCCCGTCACCACTTGACA	269
*HOXA11*	CCAATGACATACTCCTACTCTTCC	GGCTCAATGGCGTATTCTCTG	83
*FOXA2*	GTAGCCGCTCTGGGTCTTAAC	TCGTGCCCTTCCATCTTCAC	99
*PLGF*	GCCTTGTCTCCTGGGAACATT	CTCCACAAAGAAGGGCTGGT	276
*FGF7*	TCCTGCCAAGTTTGCTCTAC	CTCACTCTTATATCTCCTCCTTCC	175
*FGF10*	ACCAAGAAGGAGAACTGCCC	TTCGAGCCATAGAGTTTCCCC	131
*IGF-1*	ACCTACACAGGTGAAGATGCC	CTTGAGAGGCGCACAGTACA	286
*PGR*	CAGCCAGAGCCCACAGTACA	TGCAATCGTTTCTTCCAGCA	176
*VEGFA*	GCCTTGCCTTGCTGCTCTAC	GGTTTCTGCCCTCCTTCTGC	76
*LGR5*	CGAAACGCAAACTCCCTTCC	ATCACGAGGAAGCAAAGCGA	204
*PRLR*	CAGGTACGTACAGGGAAGCATTC	GAGTGCTTTTCATTCTGCTACTTTTTC	66
*LIF*	ACAGCCTCTTTATCCTCTATTAC	GCGATGATGCGATACAGC	151
*DKK1*	AGGTACCGTCTGTCTTCGCT	CAAGACAGGCCTTCTCCACA	180
*TCF7*	AGGAGACGACAGAGTCCCAA	ATTGAGGGCCCCAGGTTTAG	124
*FRAT2*	TACGGCGTCCTACTTACC	CCACTAACTTCTCGGCTTG	115
*WNT2*	GAACCGCCAAGGACAACAAG	TTCATCAGGGCTCTGGCATC	133
*WNT4*	GCCTTCACGGTGACTCTTCA	GCTTTGCCCTTTGTCCTGTG	248
*WNT5B*	CTGAGGACCTGGTGTACGTG	AACGCTCTTGAACTGGTCGT	164
*WNT6*	GTCGAGGCTCTTCATGGACG	GCCCGCCTCGTTGTTATGAA	85
*WNT11*	AAGGACTCGGAGCTCGTCTA	GCGGTCTGTGTAAGGGTTGT	168
*ACTB*	TCAGCAAGCAGGAGTACGAC	ACGAGGCCAATCTCATCTCG	137

## Data Availability

The data presented in this study are available in the article.

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
