# Peer review of "Roles of WNT6 in Sheep Endometrial Epithelial Cell Cycle Progression and Uterine Glands Organogenesis"

_vetsci, 2021, doi:10.3390/vetsci8120316_

Round 1

Reviewer 1 Report

Line 4: is one author name missing after 'and'

Line 61:  please provide the age of the ewes.

Line 61: If n = 10 ewes were used why are there only n = 3 per treatment? 

Line 67: From your reference [16] and this paper it is unclear to me why the ewes were slaughtered by second estrus? Why would you not do the study post implantation of embryo ? 

Table 1: please clarify what is referred to by WB/IP?

Table 1: For each Ab please include a column to provide the definition/function of each Ab.  For example what does β-catenin do?

Figure 3 and 4 legend: what are meant by NC, pEX-4-NC, pEX-4-WNT6

Line 229: Change to fertile

Overall, it is very difficult to keep track of the main message in your paper because there is so many methods, techniques and results. Can you provide a illustration which can summarise the main findings ?

Author Response

Response to Reviewer 1 Comments

Thank you very much for your review of our manuscript entitled, “Roles of WNT6 in sheep endometrial epithelial cell cycle progression and uterine glands organogenesis” (vetsci-1438271). We appreciate your concerns and suggestions, and have revised our manuscript accordingly. The manuscript has been proofread by a native English professional with science background at International Science Editing. Our responses are provided below.

Point 1: Line 4: is one author name missing after 'and'

Response 1: We thank the reviewer for pointing this out. We have corrected the mistakes in description. (Page 1, Line 5)

Point 2: Line 61:  please provide the age of the ewes.

Response 2: We are very grateful to your comments on the manuscript. We have added the the age of the ewes in the Materials and methods. (Page 2, Line 66)

Point 3: Line 61: If n = 10 ewes were used why are there only n = 3 per treatment? 

Response 3: We are sorry for the clerical error, and we have corrected the mistakes in description. (Page 2, Line 66)

Point 4: Line 67: From your reference [16] and this paper it is unclear to me why the ewes were slaughtered by second estrus? Why would you not do the study post implantation of embryo? 

Response 4: We are very grateful to your comments on the manuscript. The first estrous cycles of sheep were synchronized by progestogen-impregnated intravaginal sponges, which may affect the physiological state of endometrium between two groups. Therefore, we collected the endometrium at the second estrus (natural estrous). Estrus is also a key period when the functions of uterine glands are highly reactive in proliferation and secretion, preparing for the implantation and development of embryo. Our future research will be performed on the endometrium from post implantation. (Page 2, Line 71-76)

Point 5: Table 1: please clarify what is referred to by WB/IP?

Response 5: We thank the reviewer for pointing this out. We have corrected the mistakes. (Page 2, Line 87)

Point 6: Table 1: For each Ab please include a column to provide the definition/function of each Ab.  For example what does β-catenin do?

Response 6: We are very grateful to your comments on the manuscript. We have added the information in Table 1. (Page 2, Line 87)

Point 7: Figure 3 and 4 legend: what are meant by NC, pEX-4-NC, pEX-4-WNT6

Response 7: We thank the reviewer for pointing this out. “NC, Negative control was the siRNA with scrambled sequence absent in the sheep genome. si-WNT6, the small interfering RNA for WNT6. pEX-4-NC, Negative control of pEX-4 was the empty vector. pEX-4-WNT6, pEX-4 vector with WNT6-CDs inserted.” We have added the information in Figure 3 and 4 legend. (Page 7, Line 216-218; Page 8, Line 230-231)

Point 8: Line 229: Change to fertile

Response 8: We are very grateful to your comments on the manuscript. As our description in introduction, “uterine glands knocked out (UGKO) ewes exhibit complete infertility was attributed to recurrent pregnancy loss as the absence of specific, embryotrophic secretions from the glands.”

Point 9: Overall, it is very difficult to keep track of the main message in your paper because there is so many methods, techniques and results. Can you provide a illustration which can summarise the main findings ?

Response 9: We thank the reviewer for pointing this out. We have added the illustration in Additional file 4.

Reviewer 2 Report

The present work examined the role of WNT6 in endometrial epithelial cell cycle progression and uterine gland development in sheep. The results of this work were clearly presented, however, there are some concerns.

-Authors should include information on the number of dominant follicles present on ovaries at time of endometrial sample collection. Did HP ewes have more dominant follicles than LP ewes? This would impact estrogen concentrations, which is shown to alter endometrial gene expression. This needs to be addressed as it could have impacted transcript expression data.

-A major concern is the endometrial sample collections. Why did authors choose to only sample the left horn (line 67). There is research that highlights differential gene expression between uterine horns contralateral and ipsilateral to the dominant follicle/corpus luteum. If only the left horn was chosen and the dominant follicle is on the right ovary, mRNA expression levels could be impacted by that alone. This is particularly concerning for the LP group.

-Authors should give a brief description of methods, even if they were previously reported. An example of where this is needed is Line 124.

-Authors state that the study included 10 ewes, but only 6 ewes were used for the work (Lines 61-64). Why weren’t all 10 used?

-Grammatical errors throughout manuscript. It needs editing.

Author Response

Response to Reviewer 2 Comments

Thank you very much for your review of our manuscript entitled, “Roles of WNT6 in sheep endometrial epithelial cell cycle progression and uterine glands organogenesis” (vetsci-1438271). We appreciate your concerns and suggestions, and have revised our manuscript accordingly. The manuscript has been proofread by a native English professional with science background at International Science Editing. Our responses are provided below.

Reviewer 2

The present work examined the role of WNT6 in endometrial epithelial cell cycle progression and uterine gland development in sheep. The results of this work were clearly presented, however, there are some concerns.

Point 1: Authors should include information on the number of dominant follicles present on ovaries at time of endometrial sample collection. Did HP ewes have more dominant follicles than LP ewes? This would impact estrogen concentrations, which is shown to alter endometrial gene expression. This needs to be addressed as it could have impacted transcript expression data.

Response 1:  We thank the reviewer for pointing this out. We have not collected the information on the number of dominant follicles present on ovaries, we will take your suggestion in our future research. However, as our previous research, that there were no significant differences in estrogen levels during the estrous cycle between HP and LP Hu sheep [1].

[1] Feng X., Li F., Wang F., et al. Genome-wide differential expression profiling of mRNAs and lncRNAs associated with prolificacy in Hu sheep [J]. Biosci Rep, 2018, 38(2): BSR20171350.

Point 2:-A major concern is the endometrial sample collections. Why did authors choose to only sample the left horn (line 67). There is research that highlights differential gene expression between uterine horns contralateral and ipsilateral to the dominant follicle/corpus luteum. If only the left horn was chosen and the dominant follicle is on the right ovary, mRNA expression levels could be impacted by that alone. This is particularly concerning for the LP group.

Response 2: We are very grateful to your comments on the manuscript. In order to maintain the consistency of biological repetition, we selected the ipsilateral uterine horns of sheep for research. We will take your suggestion in our future research.

Point 3:-Authors should give a brief description of methods, even if they were previously reported. An example of where this is needed is Line 124.

Response 3: We are very grateful to your comments on the manuscript. We have added the information in Materials and methods. (Page 4, Line 133; Page 5, Line 146)

Point 4:-Authors state that the study included 10 ewes, but only 6 ewes were used for the work (Lines 61-64). Why weren’t all 10 used?

Response 4:  We are sorry for the clerical error, and we have corrected the mistakes in description. (Page 2, Line 66)

Point 5:-Grammatical errors throughout manuscript. It needs editing.

Response 5: We thank the reviewer for pointing this out. We have edited the grammatical errors throughout manuscript.

Reviewer 3 Report

In their MS, Xiaoxiao Gao and colleagues report the differences in Wnt6 pathways between Hu sheep with different proven fecundity and its association with endometrial morphology, particularly with endometrial glands development and branching. The MS contains a large amount of work, and the topic is interesting to those in the area. However, the MS needs to be submitted to language edition; as it is written, sometimes the sentences are difficult to understand.

Despite the above said, I have some concerns that will need the attention of the authors:

#1. Hu sheep breed is considered a hiperprolific breed. As in other prolific breeds, this trait is associated with a particular genetic background (related to the Boorela gene). Therefore, if the breed is well established, as I think it would be, one could expect the trait to be shared by all the females. Thus, differences in fecundity/prolificacy would derive, probably, from non-genetic influences (such as nutrition). This could be acknowledged in the MS.

Since the authors selected the animals according to the number of offspring born at lambing, it would be better to use the term "fecundity rate" or "prolificacy rate" to distinguish the two groups of animals unless they have performed genetic tests to separate animals with and without prolificacy genes.

#2. also in M&M, the population needs to be better characterized: age, number of litters produced, time from the last lambing are due
also the temporal delimitation of the study must be provided, for ewes are seasonal breeders

# 3. I have one question whose response was not clear from the reading of the MS and might not be at the focus of the current MS, but would deserve some thoughts: Does Wnt present the same functions in prepubertal gland formation and post-partum or cyclic gland adenogenesis and regeneration? Could the lower prolificacy found here result from deficient prepubertal adenogenesis of the endometrial glands, not the mature adenogenesis, and therefore be considered a critical limiting factor to the ewe fertility? How could the authors distinguish the two situations?

#4. Often the authors left undescribed procedures and/or treatments, referring them to previous work. Nonetheless, a bride description ought to be provided, particularly since they also refer to "minor modifications in some methods." for example, in lines 65-66, a sentence like this should be added: "Briefly, a CIDR vaginal sponge ( CIDR® (Controlled Internal Drug Releasing device, Pfizer Inc., New York, NY, USA), was placed for 11 days and 100 IU of PG was given at sponge removal". As it was retrieved from the cited work, please note that if PG represents prostaglandins, then the name of the drug and the supplier must be provided.

#5. Regarding the IF, Information regarding the quantification of the reaction is needed. How did you evaluate the intensity of the signal? please explain

#6. In the gene analysis, all types of controls used to validate the experiment should be mentioned.

#7. regarding the cell cycle analysis, add a short description of the interpretation of the flow cytometric analysis

#8. in the statistic analysis, have the authors tested the normal distribution of data? This information should be provided to validate the test selected for group comparisons. Even the T-test assumes the normality of data distribution.

#9.  Figure 1 presents some acronyms or abbreviations not mentioned in the caption.

#10. the reference list needs the authors' attention: in some cases, the journal name is presented in full; in others, it is abbreviated. Would you please normalize the style?

Finally, unless the authors establish a close association between the nutrition levels of the low prolificacy group (were ewes submitted to restriction diets? was their condition score below acceptable levels?) I can not see how the MS copes with the special issue topics "Animal Nutritional and Metabolic Diseases"

 However, it does match the scope of the journal Veterinary Sciences.

Author Response

Response to Reviewer 3 Comments

Thank you very much for your review of our manuscript entitled, “Roles of WNT6 in sheep endometrial epithelial cell cycle progression and uterine glands organogenesis” (vetsci-1438271). We appreciate your concerns and suggestions, and have revised our manuscript accordingly. The manuscript has been proofread by a native English professional with science background at International Science Editing. Our responses are provided below.

Reviewer 3

In their MS, Xiaoxiao Gao and colleagues report the differences in Wnt6 pathways between Hu sheep with different proven fecundity and its association with endometrial morphology, particularly with endometrial glands development and branching. The MS contains a large amount of work, and the topic is interesting to those in the area. However, the MS needs to be submitted to language edition; as it is written, sometimes the sentences are difficult to understand.

Response:We are very grateful to your comments on the manuscript. The manuscript has been proofread by a native English professional with science background at International Science Editing.

Despite the above said, I have some concerns that will need the attention of the authors:

Point 1:#1. Hu sheep breed is considered a hiperprolific breed. As in other prolific breeds, this trait is associated with a particular genetic background (related to the Boorela gene). Therefore, if the breed is well established, as I think it would be, one could expect the trait to be shared by all the females. Thus, differences in fecundity/prolificacy would derive, probably, from non-genetic influences (such as nutrition). This could be acknowledged in the MS.

Response 1:We are very grateful to your comments on the manuscript. We have added the information in Information. (Page 1, Line 31)

Point 1-1: Since the authors selected the animals according to the number of offspring born at lambing, it would be better to use the term "fecundity rate" or "prolificacy rate" to distinguish the two groups of animals unless they have performed genetic tests to separate animals with and without prolificacy genes.

Response 1-1:We thank the reviewer for pointing this out. We have corrected the mistakes in description.

Point 2: #2. also in M&M, the population needs to be better characterized: age, number of litters produced, time from the last lambing are due, also the temporal delimitation of the study must be provided, for ewes are seasonal breeders.

Response 2: We are very grateful to your comments on the manuscript. We have added the information in M&M. (Page 2, Line 66-68)

Point 3: # 3. I have one question whose response was not clear from the reading of the MS and might not be at the focus of the current MS, but would deserve some thoughts: Does Wnt present the same functions in prepubertal gland formation and post-partum or cyclic gland adenogenesis and regeneration? Could the lower prolificacy found here result from deficient prepubertal adenogenesis of the endometrial glands, not the mature adenogenesis, and therefore be considered a critical limiting factor to the ewe fertility? How could the authors distinguish the two situations?

Response 3: We thank the reviewer for pointing this out. As reported, the activation of Wnt/β-catenin signalling in proliferation period is a prerequisite for proper decidualization and correct invasion of trophoblast into the maternal endometrium. We will explore the role of Wnt in prepubertal gland formation in future study.

Van Der Horst P. H., Wang Y., Van Der Zee M., et al. Interaction between sex hormones and WNT/beta-catenin signal transduction in endometrial physiology and disease [J]. Mol Cell Endocrinol, 2012, 358(2): 176-84.

Point 4: #4. Often the authors left undescribed procedures and/or treatments, referring them to previous work. Nonetheless, a bride description ought to be provided, particularly since they also refer to "minor modifications in some methods." for example, in lines 65-66, a sentence like this should be added: "Briefly, a CIDR vaginal sponge ( CIDR® (Controlled Internal Drug Releasing device, Pfizer Inc., New York, NY, USA), was placed for 11 days and 100 IU of PG was given at sponge removal". As it was retrieved from the cited work, please note that if PG represents prostaglandins, then the name of the drug and the supplier must be provided.

Response 4: We are very grateful to your comments on the manuscript. We have added the information in M&M. (Page 2, Line 71-74)

Point 5: #5. Regarding the IF, Information regarding the quantification of the reaction is needed. How did you evaluate the intensity of the signal? please explain

Response 5: We thank the reviewer for pointing this out. Uterine glands were displayed red fluorescence, and sections were taken to count the uterine glands using a Nikon microscope (Nikon Inc., Tokyo, Japan). The density of uterine glands was determined at high magnification in each of the five random sampling areas.

Point 6: #6. In the gene analysis, all types of controls used to validate the experiment should be mentioned.

Response 6: We are very grateful to your comments on the manuscript. We have added the information. (Page 3, Line 114; Page 4, Line 138-140; Figure 3-4 legends)

Point 7: #7. regarding the cell cycle analysis, add a short description of the interpretation of the flow cytometric analysis

Response 7: We thank the reviewer for pointing this out. We have added the information. (Page 5, Line 145-151)

Point 8: #8. in the statistic analysis, have the authors tested the normal distribution of data? This information should be provided to validate the test selected for group comparisons. Even the T-test assumes the normality of data distribution.

Response 8: We are very grateful to your comments on the manuscript. We have added the information. (Page 5, Line 145-151)

Point 9: #9.  Figure 1 presents some acronyms or abbreviations not mentioned in the caption.

Response 9: We thank the reviewer for pointing this out. We have added the information. (Page 5-6, Line 174-182)

Point 10: #10. the reference list needs the authors' attention: in some cases, the journal name is presented in full; in others, it is abbreviated. Would you please normalize the style?

Response 10:  We thank the reviewer for pointing this out. We have normalized the style of reference.

Finally, unless the authors establish a close association between the nutrition levels of the low prolificacy group (were ewes submitted to restriction diets? was their condition score below acceptable levels?) I can not see how the MS copes with the special issue topics "Animal Nutritional and Metabolic Diseases"However, it does match the scope of the journal Veterinary Sciences.

Response: We are very grateful to your comments on the manuscript.

Round 2

Reviewer 3 Report

In the resubmitted MS, the authors attend to most of my concerns. Even though, there are still some issues that need to be revised, to improve the MS quality.

  • Line 55 - the references provided for the sentence "in humans and bovines" both address to human. Authors should provide a reference supporting the mention to bovines.
  • In M&M, some of the methodology used should be a little more explained, for those that would like to replicate the study, particularly if changes to the original method were introduced (even if minor).

for the IF, for examples, information still amiss include: 

-Were samples for IF  fixed in any compound? or you used cryostat sections?
- Were samples embedded in parafin or a resin?
- What was the basic procedure to get a section?
- What was the section thickness used in here?
- How did you quenched the endogenous autofluorescence?
- some of the antibodies used are not validated for sheep - How did you validate its use in your species? - add the information, please

- provide the wavelenght used for the fluorochrome

- Details on the time and temperature of incubation must be provided

  • Table 1 - the function was not presented in the last column ! in some cases, not even the description was provided.
  • please provide the mean for each abbreviation used.
    if not within the text, as it may be currently applied in the area, at least at the beginning or the end of the MS, from novice readers
  • Figure 6 - 

    it is not clear, to me which abbreviation represents the inhibition and th overexpression condition in the images and graphs

    correct the wording of the caption so it becomes more clear to the reader

  • Additional minor suggestions were introduced in the commented file, in attach

Author Response

Response to Reviewer 3 Comments

Thank you very much for your review of our manuscript entitled, “Roles of WNT6 in sheep endometrial epithelial cell cycle progression and uterine glands organogenesis” (vetsci-1438271). We appreciate your concerns and suggestions, and have revised our manuscript accordingly. Our responses are provided below.

Reviewer 3

In the resubmitted MS, the authors attend to most of my concerns. Even though, there are still some issues that need to be revised, to improve the MS quality.

Point 1: Line 55 - the references provided for the sentence "in humans and bovines" both address to human. Authors should provide a reference supporting the mention to bovines.

Response 1: We thank the reviewer for pointing this out. We have corrected the mistakes in description. (Page 2, Line 55)

Point 2: In M&M, some of the methodology used should be a little more explained, for those that would like to replicate the study, particularly if changes to the original method were introduced (even if minor). for the IF, for examples, information still amiss include:

-Were samples for IF fixed in any compound? or you used cryostat sections?

- Were samples embedded in parafin or a resin?

- What was the basic procedure to get a section?

- What was the section thickness used in here?

- How did you quenched the endogenous autofluorescence?

- some of the antibodies used are not validated for sheep - How did you validate its use in your species? - add the information, please

- provide the wavelenght used for the fluorochrome

- Details on the time and temperature of incubation must be provided

Response 2: Thank you for your kind comments. We have added the information in M&M. (Page 2-3, Line 91-104).

The sequence homology of the target sequence for PCNA, WNT6, β-catenin, Cyclin D1, c-MYC, FOXA2, and Cytokeratin was more than 85% with the antibodies cross-reactivity species (Human). The results for sequence homology of the target sequence for antibodies are as following (www.uniprot.org/blast/). (Page 4, Line 110).

Point 3: Table 1 - the function was not presented in the last column ! in some cases, not even the description was provided.

please provide the mean for each abbreviation used. if not within the text, as it may be currently applied in the area, at least at the beginning or the end of the MS, from novice readers

Response 3: We thank the reviewer for pointing this out. We have added the information in Table 1. (Page 3, Line 107).

Point 4: Figure 6 - it is not clear, to me which abbreviation represents the inhibition and the overexpression condition in the images and graphs, correct the wording of the caption so it becomes more clear to the reader

Response 4: We thank the reviewer for pointing this out. We have corrected the mistakes in description. (Page 12, Line 294-297).

Point 5: Additional minor suggestions were introduced in the commented file, in attach

Response 5: Thank you for your kind comments. We have corrected the mistakes in manuscript.

Round 3

Reviewer 3 Report

In the resubmit MS, the authors answered my concerns in full. I have found only minor issues that can be trusted the authors for correction.

I recommend this MS for publication

Author Response

Reviewer 3-3

Point 1: In the resubmit MS, the authors answered my concerns in full. I have found only minor issues that can be trusted the authors for correction.

Response 1: We thank the reviewer for pointing this out. We have corrected the mistakes in MS.

This manuscript is a resubmission of an earlier submission. The following is a list of the peer review reports and author responses from that submission.